# Current Perspectives on Rehabilitation Following Return of Spontaneous Circulation After Sudden Cardiac Arrest: A Narrative Review

**DOI:** 10.3390/healthcare13151865

**Published:** 2025-07-30

**Authors:** Kamil Salwa, Karol Kaziród-Wolski, Dorota Rębak, Janusz Sielski

**Affiliations:** 1Intensive Cardiac Care Unit, Świetokrzyskie Cardiology Center, 25-736 Kielce, Poland; kamil.salwa@gmail.com (K.S.); karol.kazirod-wolski@ujk.edu.pl (K.K.-W.); 2Institute of Medical Sciences, Collegium Medicum, Jan Kochanowski University in Kielce, 25-317 Kielce, Poland; 3Institute of Health Sciences, Collegium Medicum, Jan Kochanowski University in Kielce, 25-317 Kielce, Poland; dorota.rebak@ujk.edu.pl

**Keywords:** post-cardiac arrest rehabilitation, ROSC, neurocognitive recovery, early mobilization, SCA, functional recovery

## Abstract

**Background/Objectives:** Sudden cardiac arrest (SCA) is a major global health concern with high mortality despite advances in resuscitation techniques. Achieving return of spontaneous circulation (ROSC) represents merely the initial step in the extensive rehabilitation journey. This review highlights the critical role of structured, multidisciplinary rehabilitation following ROSC, emphasizing the necessity of integrated physiotherapy, neurocognitive therapy, and psychosocial support to enhance quality of life and societal reintegration in survivors. **Methods:** This narrative review analyzed peer-reviewed literature from 2020–2025, sourced from databases such as PubMed, Scopus, Web of Science, and Google Scholar. Emphasis was on clinical trials, expert guidelines (e.g., European Resuscitation Council 2021, American Heart Association 2020), and high-impact journals, with systematic thematic analysis across rehabilitation phases. **Results:** The review confirms rehabilitation as essential in addressing Intensive Care Unit–acquired weakness, cognitive impairment, and post-intensive care syndrome. Early rehabilitation (0–7 days post-ROSC), focusing on parameter-guided mobilization and cognitive stimulation, significantly improves functional outcomes. Structured interdisciplinary interventions encompassing cardiopulmonary, neuromuscular, and cognitive domains effectively mitigate long-term disability, facilitating return to daily activities and employment. However, access disparities and insufficient randomized controlled trials limit evidence-based standardization. **Discussion:** Optimal recovery after SCA necessitates early and continuous interdisciplinary engagement, tailored to individual physiological and cognitive profiles. Persistent cognitive fatigue, executive dysfunction, and emotional instability remain significant barriers, underscoring the need for holistic and sustained rehabilitative approaches. **Conclusions:** Comprehensive, individualized rehabilitation following cardiac arrest is not supplementary but fundamental to meaningful recovery. Emphasizing early mobilization, neurocognitive therapy, family involvement, and structured social reintegration pathways is crucial. Addressing healthcare disparities and investing in rigorous randomized trials are imperative to achieving standardized, equitable, and outcome-oriented rehabilitation services globally.

## 1. Introduction

SCA remains a major cause of mortality in Europe. The American Heart Association defines it as an abrupt stop in cardiac function, irrespective of prior heart disease diagnosis [1].

Although Europe reports fewer cardiac arrests than other regions, OHCA affects 67–170 per 100,000 people annually, while in-hospital arrests occur in 1.5–2.8 per 1000 patients yearly [2].

Basic Life Support (BLS) is now central to public education, yet CPR is performed by trained responders in just 50–60% of OHCA cases. Despite dispatcher-assisted CPR in most countries, barriers delay emergency calls. Early CPR and prompt defibrillation are key to survival [3].

Public AED deployment is expanding, as defibrillation within five minutes can boost survival to 70% [4].

Advanced Life Support (ALS) implementation, now standard in healthcare, enhances survival, especially to discharge. Resuscitation continues after ROSC, transitioning to structured post-resuscitation care [5].

In 2015, the ERC issued unified guidelines, developed through global consensus including contributors from five continents [6].

These guidelines address post-arrest care: ensuring oxygenation, hemodynamic stability, coronary perfusion, TTM, seizure control, prognosis, rehabilitation, and long-term outcomes. Post-cardiac arrest syndrome management is emphasized. All ROSC patients need immediate, evidence-based intensive care [7].

Founded in 1989, the ERC leads in resuscitation protocols and education across Europe, integrating research into guidelines and promoting cardiac arrest awareness. As part of ILCOR, it helps shape global CoSTR, with updates published in Resuscitation [7].

### Objectives of the Review

This review outlines current evidence-based rehabilitation strategies after ROSC following sudden cardiac arrest. Beyond survival, it stresses the need for structured, multidisciplinary care to restore function, cognition, and psychosocial well-being. By analyzing recent literature and guidelines, it addresses care gaps, promotes individualized rehabilitation, and underscores the roles of physiotherapy, neurocognitive therapy, and family support throughout recovery. The goal is to guide clinical decisions, standardize care, and advance outcome-focused rehabilitation globally.

## 2. Materials and Methods

This narrative review used a structured search of peer-reviewed literature from 2020–2025 across PubMed, Scopus, Web of Science, and Google Scholar, focusing on top journals like Resuscitation, Circulation, and Critical Care Medicine. Studies on rehabilitation after ROSC were selected using terms such as “ROSC rehabilitation,” “post-cardiac arrest recovery,” “early ICU mobilization,” and “neurocognitive therapy post-arrest.” The inclusion was based on relevance, rigor, and fit with SCA recovery. Clinical trials, cohort studies, expert guidelines (e.g., ERC 2021, AHA 2020), and structured reviews were prioritized. Reference tracing ensured coverage of key guideline-cited evidence. Publications were thematically analyzed by rehabilitation phase to support the review’s goal of presenting current, integrated, multidisciplinary strategies. Recent high-quality findings were incorporated. The detailed inclusion and exclusion criteria used in the review are shown in Table 1. In addition, the process of reviewing and obtaining publications for review is shown by the algorithm in Figure 1.

## 3. Results

### 3.1. Epidemiology

Cardiovascular diseases, especially CAD and AMI, remain the top global causes of death, increasingly affecting both high- and middle-income nations. These often precede SCA–an abrupt cardiac arrest with minimal warning and poor survival, even with rapid care. Despite emergency care advances, OHCA incidence remains 67–170 per 100,000 annually, with hospital discharge survival below 10% in many regions [8].

IHCA, though in controlled settings, shows variable rates (1.5–2.8 per 1000), shaped by patient complexity and system readiness [9].

SCA outcomes depend on multiple factors. Structural heart diseases like ischemic cardiomyopathy and valvular defects raise the risk of arrest and mortality. A Scandinavian registry linked left-sided valve lesions to worse OHCA outcomes, stressing early structural assessment during rehab [9].

SCA also shows circadian and seasonal variation, indicating neurohumoral and autonomic influences. ROSC, though essential, is not enough; rates vary widely, averaging 20–40% in efficient systems [10].

Critically, ROSC does not ensure neurologic recovery. Outcomes rely on prompt ALS, neuroprotection, and rehab—often inconsistently applied [8,10].

Moreover, new directions in cardiac arrest epidemiology emphasize the integration of real-time data analytics, risk modeling, and predictive algorithms, including artificial intelligence-based systems, to improve early recognition, triage, and post-arrest stratification. Emerging literature also calls for the refinement of international resuscitation registries and reporting standards to unify data on IHCA, OHCA, and ROSC in a way that reflects not only survival, but also quality-adjusted life years (QALYs) and social reintegration outcomes [11].

Robust epidemiological insight is essential to identify risks, guide acute responses, and shape rehabilitation. Future strategies must stress early coronary syndrome detection, access to resuscitation, and fair rehab implementation [12].

### 3.2. The Fourth Link in the Survival Chain

Historically, achieving return of spontaneous circulation (ROSC) was regarded as the endpoint of resuscitation, with minimal attention to survivors’ long-term outcomes. The recent formal addition of “Recovery” to the chain of survival signifies a critical paradigm shift, expanding the focus of post-arrest care toward rehabilitation. Notably, survivors of sudden cardiac arrest (SCA) frequently exhibit a broader spectrum of deficits than typical ICU patients. In response, a specialized ROSC-Rehabilitation (ROSC-Rehab) team has been proposed to address these complex needs. This multidisciplinary unit, closely integrated with critical care, would aim to restore function and autonomy through coordinated, phase-specific rehabilitation. Initial management should reflect protocols for the critically ill, followed by direct transition into the ROSC-Rehab pathway upon ICU discharge, regardless of arrest etiology. This approach consolidates multiple specialties—including cardiac and neurological rehabilitation—into a unified framework, minimizing care fragmentation. Importantly, it may also catalyze the emergence of clinicians with dedicated expertise in post-arrest recovery. As patients return home, the ROSC-Rehab team, supported by digital tools, should provide continuous monitoring, reinforce adherence, and assist caregivers [2].

Furthermore, rehabilitation must support reintegration into daily life and employment, while promoting lasting lifestyle changes. Given the benefits of early mobilization and existing ICU rehab structures, expanding to a comprehensive ROSC-Rehab model is both timely and warranted. Nevertheless, resource-limited settings face significant barriers, chiefly due to workforce constraints. In such environments, task-sharing and task-shifting may prove vital for sustaining rehabilitation services. Ultimately, global implementation of ROSC-Rehab will demand gradual standardization, robust research, and cultural adaptation to ensure that all SCA survivors receive equitable, effective care [13].

### 3.3. Post Cardiac Arrest Syndrome-PCAS

Survival after ROSC is frequently complicated by Post-Cardiac Arrest Syndrome (PCAS), a complex condition stemming from global ischemia–reperfusion injury and a dysregulated inflammatory response, as emphasized in current international guidelines [2,14].

PCAS encompasses four principal domains: hypoxic–ischemic brain injury, myocardial dysfunction, systemic reperfusion response, and the precipitating cause of arrest. The cerebral cortex is highly vulnerable; within minutes of arrest, neuronal energy failure and calcium overload trigger apoptotic and necrotic cascades. Reperfusion exacerbates injury via oxidative stress and pro-inflammatory mediators, leading to coma, seizures, or lasting cognitive decline [15].

Among neuroprotective strategies, targeted temperature management (TTM) remains the most studied, though its efficacy is still under debate [16].

Myocardial dysfunction, particularly myocardial stunning, frequently follows ROSC and manifests as transient reductions in output, hypotension, and arrhythmias. Although reversible, it complicates early hemodynamic stabilization and impedes rehabilitation efforts [14].

Simultaneously, systemic reperfusion elicits an inflammatory response resembling sepsis, with endothelial dysfunction, vascular leakage, and mitochondrial injury. Elevated cytokines, notably IL-6 and TNF-α, drive multi-organ dysfunction, affecting renal, hepatic, and pulmonary systems [15].

Therefore, mitigating this cascade is central to post-arrest therapy. Equally critical is identifying and treating the primary etiology—whether ischemic, thromboembolic, hypoxic, or metabolic—as failure may compromise systemic recovery [16].

Each PCAS domain significantly influences prognosis and rehabilitation. Neurologic deficits delay mobilization; cardiovascular instability hinders physical therapy. Early, comprehensive evaluation—especially of neurologic and cardiopulmonary function—is essential to guide individualized rehabilitation and maximize recovery in survivors [15].

### 3.4. Early Neurological Assessment Significance

Neurological injury constitutes a principal determinant of long-term prognosis after SCA resuscitation, often delineating the ceiling of functional recovery and life quality [17].

The immediate post-ROSC period entails ongoing neuronal damage driven by ischemic, metabolic, and inflammatory cascades. In this vulnerable phase, targeted temperature management (TTM)—formerly therapeutic hypothermia—has emerged as a critical intervention to mitigate secondary brain injury [18].

As highlighted by Feitosa-Filho et al., early induction of mild hypothermia notably improves neurologic outcomes and survival, especially in comatose patients with initial rhythms of ventricular fibrillation or pulseless ventricular tachycardia. Cooling to 32–34 °C for 12–24 h reduces cerebral metabolic demand and disrupts key injurious mechanisms, including oxidative stress, calcium overload, and inflammatory activation [19].

This intervention demonstrates substantial efficacy, with a number needed to treat (NNT) of six, underscoring its clinical impact [20].

Furthermore, both external and intravascular cooling techniques have proven feasible, with particular emphasis placed on controlled rewarming to avoid complications such as intracranial hypertension or hemodynamic instability. Importantly, TTM also facilitates physiological stabilization, enhancing the reliability of early neurologic assessments [21].

Consequently, improved prognostic clarity enables earlier rehabilitation planning. TTM should therefore be viewed not in isolation, but as an integral element of post-resuscitation care—linking acute management with personalized recovery trajectories [19,22].

### 3.5. Metabolic Profiling and Secondary Risk in Cardiac Arrest Survivors

Although the SCARF study does not employ advanced biochemical profiling, it offers clinical relevance by identifying functional and behavioral markers—such as fatigue, inactivity, and psychological burden—as early indicators of vulnerability in SCA survivors [23].

These surrogate indicators help capture metabolic dysregulation post-ischemia-reperfusion, suggesting that clustered symptoms may reflect underlying systemic strain [24].

Importantly, fatigue is reframed as a multifactorial manifestation involving residual cerebral injury, cognitive load, autonomic imbalance, chronic inflammation, and reduced cardiac output. Collectively, this symptom constellation may signify persistent metabolic instability, comparable to patterns observed in critical illness survivors [25,26,27].

Findings from SCARF support this interpretation. While changes in fatigue scales (MFIS, MFI-20) were modest, participants achieved measurable gains in objective function (e.g., chair-stand, six-minute walk) and in WHO-defined activity and participation metrics [13].

These results imply that targeted, multidimensional rehabilitation can partially reverse the functional manifestations of metabolic stress [28].

Crucially, high baseline fatigue correlated with program dropout, indicating that low metabolic reserve may impair engagement and adherence. This observation should guide future trial design and patient stratification based on early functional–metabolic risk. Collectively, SCARF endorses a shift toward a functional–metabolic framework in post-SCA care, where symptoms such as fatigue are recognized not as secondary complaints, but as key markers of biological stress demanding early, multidisciplinary intervention [13].

### 3.6. Physiotherapy and Early Rehabilitation Interventions in ROSC

Figure 2 illustrates the progression in the rehabilitation process following ROSC.

#### 3.6.1. Acute Phase (0–72 h)

The immediate post-ROSC phase constitutes a physiologically unstable yet therapeutically pivotal window. While traditional care emphasizes hemodynamic stabilization and neuroprotection, growing evidence highlights the value of initiating individualized early rehabilitation to support long-term functional and neurological outcomes. As Feitosa-Filho et al. note, targeted temperature management during this phase modulates systemic stress, establishing a more favorable physiological environment for intervention [19].

Building on this concept, recent studies by Christensen et al. and Joshi et al. support introducing low-intensity therapies—such as passive mobilization, upright positioning, and assisted breathing—within 48–72 h post-ROSC. These measures have been associated with enhanced physical endurance and improved cognitive responsiveness during coma emergence, with digital tools refining delivery [13,29].

Furthermore, mobility tracking using wearables and computer vision, as demonstrated by Siegel et al. and Davoudi et al., offers predictive insight into neurologic outcomes and enables real-time adaptation of mobilization protocols [30,31]. Ren et al. add that dynamic ICU scoring can guide the titration of rehabilitation intensity according to physiological stability. In parallel, nutrition is increasingly recognized as a modifiable factor [32].

Some researchers report that controlled underfeeding in early critical illness may foster a more adaptive metabolic state, underscoring the need to align nutritional support with rehabilitation goals. This view is consistent with current AHA and ERC guidelines, which recommend a comprehensive assessment of neurologic, hemodynamic, and metabolic readiness prior to initiating mobilization. Despite inherent risks—particularly in sedated or unstable patients—emerging data, including from the Contreras model and neurocognitive monitoring, support patient-specific mobilization as a strategy to mitigate complications such as delirium and prolonged unconsciousness [15,33].

#### 3.6.2. Early Phase (3–7 Days)

The period between 72 h and the end of the first post-ROSC week marks a critical phase in recovery, offering a strategic window to begin structured rehabilitation that mitigates neuromuscular decline, pressure injuries, and functional loss. As sedation and neuroprotective protocols are withdrawn, clinical focus transitions from passive stabilization to targeted therapeutic activation. Consciousness and neuromuscular responsiveness gradually return, allowing for individualized rehabilitation planning. The ROCK study emphasizes that initiating patient-specific strategies during this stage is key to preserving function and preventing secondary complications [29].

During this subacute period, systematic evaluations of motor function and cognition inform the initiation of targeted interventions, including bed-to-chair transfers, upright positioning, core and postural training, and low-resistance strengthening. These therapies help counteract critical illness myopathy, which is closely linked to delayed recovery and long-term disability [34].

Strategically introducing rehabilitation during this stage can enhance mobilization tolerance, accelerate early gains, and serve as a functional bridge to long-term recovery. While pressure injury prevention is standard in critical care, it gains heightened relevance here due to impaired sensory–motor responsiveness. Dynamic surfaces, frequent repositioning, and friction-reducing technologies are essential for maintaining tissue integrity. Early functional strategies must move beyond complication prevention to actively promote recovery. Studies by Joshi et al. and Shen et al. support the integration of passive and active range-of-motion exercises into neurocognitive stimulation frameworks to maintain muscle tone and posture [13].

This reflects a shift in early rehabilitation philosophy—from passive prevention to proactive readiness for functional independence [1,13,35].

Mion and Keeble underscore the psychosocial dimension, demonstrating that family presence and familiar stimuli facilitate cognitive re-engagement. This phase serves as a bridge between physiological stabilization and early reintegration, guided by markers such as improving oxygenation, EEG patterns, and voluntary movement [36].

Košir and Harding et al. support adjusting therapy intensity according to these markers, ensuring alignment with neurologic recovery. Incorporating psychosocial support with physical activity enhances engagement and emotional resilience, providing a foundation for comprehensive rehabilitation [37].

Despite evidence variability, consensus is growing that days 3–7 post-ROSC represent a therapeutic opportunity. Rather than a passive observation period, this phase should be leveraged for interdisciplinary action to reduce post-intensive care syndrome (PICS), support ventilator weaning, and initiate functional recovery. Coordinated input from rehabilitation, critical care, neurology, and psychosocial services is essential to optimize long-term outcomes [36].

#### 3.6.3. Functional Phase

Beyond the first post-ROSC week, rehabilitation shifts from passive mobilization to structured, goal-oriented recovery. At this stage, cumulative effects of immobility emerge—marked by neuromuscular deconditioning, impaired posture, and psychomotor deficits—particularly in those confined to prolonged supine positioning. Physiotherapy targets restoration of anti-gravity posture, upright tolerance, and motor autonomy. The ROCK trial confirmed that initiating multidisciplinary rehabilitation shortly after ICU discharge, and continuing beyond day 7, significantly improves independence in ADLs [29].

Early interventions—bridging, segmental rolling, assisted sitting—reactivate proprioceptive and vestibular systems, preparing patients for upright mobility [19].

Shen et al. demonstrated that transitioning from bed rest to mobility requires structured postural progression and paced endurance. Overexertion during this phase increases risk for fatigue and orthostatic intolerance, underscoring the need for titrated progression [35].

Functional recovery must align with psychosocial context. Mion and Keeble emphasized integrating communication, social stimuli, and shared goals to enhance orientation and emotional resilience, particularly in patients with delayed awakening or hypoxic injury [36].

Joshi et al., through the SCARF study, showed that structured post–day 7 rehabilitation—including upright tolerance, neuromotor reconditioning, and gait retraining—yields gains in physical function and mental wellbeing, especially in those recovering from full dependency [13].

In line with ERC and AHA guidance, this phase requires individualized, interdisciplinary input. Recovery must be assessed not only by ambulation but by returning to complex daily functions, work, and social participation—outcomes achievable only through sustained, coordinated rehabilitation [36].

#### 3.6.4. Long-Term Rehabilitation Phase

As survival after cardiac arrest improves, the emphasis increasingly shifts toward long-term outcomes and societal reintegration. The late rehabilitation phase—initiated post-discharge or after stabilization of major deficits—prioritizes functional independence, role resumption, and psychosocial recovery. Sustaining gains beyond hospitalization requires targeted interventions: individualized home-based programs, community services, and intensive residential models. Return-to-work (RTW) is now a key composite indicator of recovery. In their longitudinal analysis of the ROCK trial, Christensen et al. found that despite resolved neurological deficits, patients frequently faced cognitive fatigue, executive dysfunction, and neuropsychiatric symptoms hindering work reentry. Effective RTW demands tailored pathways integrating occupational therapy, employer coordination, and gradual reintegration strategies [29].

Complementary to center-based care, Joshi et al. demonstrated the feasibility of home-based rehabilitation, incorporating virtual physiotherapy, caregiver-assisted exercises, and remote monitoring. This approach improved adherence and outcomes, particularly for those with psychological or logistical barriers to in-person care [13].

Mion and Keeble explored intensive rehabilitation camps offering structured therapy, neurocognitive training, and peer support. Participants reported enhanced motivation, self-efficacy, and emotional resilience—outcomes often neglected in outpatient settings. These camps serve as a potent adjunct to home and community models [38].

Yet, as Kristensen et al. note, many survivors experience residual deficits in decision-making, emotional regulation, and executive function, undetectable in routine follow-up but with major implications for autonomy and occupational function. Long-term rehabilitation must therefore include neuropsychological assessment and targeted cognitive–behavioral support [34].

ERC 2021 guidelines further emphasize that PICS—a multidimensional syndrome involving cognitive, emotional, and physical impairments—may persist for years. Rehabilitation systems must offer prolonged, flexible engagement with responsive escalation pathways to address stagnation or clinical decline [2].

In sum, effective long-term recovery post-ROSC requires abandoning time-limited models in favor of a personalized, multidisciplinary continuum—whether via RTW programs, home-based care, or residential camps—focused on restoring autonomy, function, and psychosocial identity [29].

### 3.7. The Role of the Interdisciplinary Team

Effective post-cardiac arrest rehabilitation necessitates a dynamic, patient-centered, interdisciplinary model. Given the complexity of PCAS—including cardiovascular, neurological, cognitive, and musculoskeletal impairments—no single discipline can ensure comprehensive recovery. Optimal outcomes rely on coordinated input from cardiology, neurology, physiotherapy, neuropsychology, and occupational therapy, each contributing domain-specific expertise across the care continuum.

Mion and Keeble underscore that early integration—ideally within the first ICU week—is crucial for downstream recovery, guided by principles of continuity, coordination, and contextual relevance. The cardiologist manages hemodynamics, cardiac function, pharmacotherapy, and readiness for mobilization, tailoring activity plans to individual risk profiles [36].

In parallel, the neurologist directs neuroprognostication through imaging, EEG analysis, and cognitive assessment, facilitating the timely initiation of neurorehabilitation and coordination with cognitive specialists [39].

The physiotherapist leads physical restoration, commencing within 72 h post-stabilization. Focus areas include joint mobility, anti-gravity transition, and gradual progression to gait training and cardiopulmonary reconditioning, forming the basis for daily reintegration [39].

The neuropsychologist addresses persistent cognitive sequelae—executive dysfunction, memory loss, and emotional dysregulation—common barriers to return-to-work and independence. Targeted assessments and therapies restore decision-making, self-regulation, and adaptability [40].

Occupational therapists (OTs) translate clinical gains into functional autonomy. By facilitating ADLs, vocational tasks, assistive device training, and environmental adaptation, OTs enable patients to resume meaningful societal roles [36].

Critically, the team’s strength lies in its cohesion: shared goals, regular interdisciplinary reviews, and transparent communication ensure aligned, responsive care. This model counters fragmented systems that, as Gräsner et al. highlight, hinder rehabilitation continuity across Europe [3].

In conclusion, interdisciplinary care is not optional but essential. Each discipline contributes uniquely, yet collaboratively, to restoring autonomy, minimizing disability, and supporting full reintegration. This unified framework underpins successful, long-term recovery after cardiac arrest [39].

### 3.8. The Importance of Assessing Cognitive Function and Activities of Daily Living in Cardiac Arrest Survivors

Despite improved survival after SCA, effective rehabilitation must extend beyond physical recovery to include structured assessment of cognitive function and activities of daily living (ADLs)—key predictors of long-term independence and reintegration. Many survivors, though free of gross deficits, experience subtle impairments in attention, memory, and executive function that often remain undetected yet markedly reduce autonomy and occupational success. Kristensen et al. found that over 50% of OHCA survivors exhibited impairments in task initiation, sequencing, and completion—despite preserved motor function—using AMPS and COPM assessments. These deficits were strongly linked to lower RTW rates and hindered community reintegration, reinforcing the need for comprehensive cognitive and ADL evaluation post-ROSC [34].

Similarly, the ROCK trial by Christensen et al. showed that RTW depended as much on cognitive resilience—particularly planning and multitasking—as on cardiopulmonary recovery. Hypoxic–ischemic injury frequently disrupts these domains, even when standard neurologic exams appear normal, warranting formal neuropsychological testing (e.g., MoCA, TMT) to guide intervention [29].

Targeted cognitive rehabilitation must align with identified deficits. The SCARF study by Joshi et al. demonstrated that patients with low MoCA scores improved most through integrated cognitive–behavioral and occupational therapy, particularly when training was embedded in real-world tasks (e.g., budgeting, navigation, meal prep), translating into greater functional independence [13].

Feitosa-Filho et al. further noted that early attentional deficits—especially following therapeutic hypothermia—predicted delayed ADL recovery. Staggered emergence from sedation complicates assessment, requiring phased rehabilitation with adaptive intensity and timing [19].

Combining formal cognitive tests with ADL observation offers a robust strategy for identifying at-risk survivors. Occupational therapists, neuropsychologists, and physiatrists play central roles, informing discharge planning and directing post-acute care—whether through community rehab, support services, or caregiver engagement [34].

In sum, hidden cognitive and functional impairments are common despite apparent recovery and, if overlooked, risk chronic disability and social exclusion. Integrating systematic cognitive-ADL assessment into standard post-ROSC care is essential for guiding targeted, multidisciplinary rehabilitation and ensuring safe, meaningful reintegration [34].

### 3.9. Education and Support of the Patient’s Family After Return of Spontaneous Circulation (ROSC)

Recovery after SCA marks not only a complex physiological process for the patient but also an emotionally taxing journey for the family. Within post-ROSC care, structured family education and psychosocial support are now essential. Families often serve as caregivers and decision-makers, yet they remain underprepared and unsupported—especially during ICU admission. Mion and Keeble stress that individualized, early education reduces emotional burden and aligns expectations with rehabilitation goals. Effective communication must go beyond clinical updates to include visual tools, written materials, and psychosocial counseling tailored to the family’s readiness [36].

Similarly, the SCARF study showed improved home-based rehabilitation outcomes when caregivers were involved in goal-setting and trained in basic interventions. Caregiver confidence directly correlated with patient adherence, underscoring the family’s role as an extension of the clinical team [13].

The ROCK trial by Christensen et al. further validated family-focused modules, which addressed cognitive sequelae, reintegration challenges, and stress management. These interventions reduced caregiver burden (Zarit scores) and supported community reintegration, affirming the therapeutic value of psychoeducation [29].

Beyond information, emotional support remains critical. Caregivers face elevated risks of PICS-F, including anxiety, depression, and PTSD. Nolan et al. advocate for structured psychological screening, mental health referrals, respite access, and mentoring—all vital to caregiver resilience and sustainable recovery [2].

Giuvară et al. emphasize realistic outcome framing to prevent disengagement from mismatched expectations. Clear, evidence-based communication on possible delays, plateaus, and residual deficits fosters sustained, informed participation [39].

In line with ERC recommendations, discharge planning should include formal family meetings, home adaptation strategies, and follow-up involvement to ensure smooth transitions from the ICU to home. Families, when equipped with knowledge, support, and structured roles, become essential partners in rehabilitation. Their engagement enhances outcomes, reduces secondary trauma, and supports durable recovery [39].

### 3.10. Clinical Barriers and Complications Affecting Early Rehabilitation After ROSC

Early rehabilitation after ROSC is often limited by concurrent interventions and post-resuscitation pathophysiology, necessitating individualized planning and interdisciplinary coordination.

#### 3.10.1. Urgent Percutaneous Coronary Intervention (PCI)

In STEMI patients, early PCI is standard but may delay mobilization due to vascular sheath presence and bleeding risk. Dual antiplatelet therapy (DAPT), often combined with anticoagulation, further elevates hemorrhagic risk. Thus, in-bed activity and avoidance of Valsalva maneuvers are advised during the first 24–48 h [41,42,43].

#### 3.10.2. Hemodynamic Instability and Vasoactive Support

Myocardial dysfunction may induce cardiogenic shock, requiring vasopressors and invasive monitoring. Hypotension, elevated lactate, and impaired perfusion limit exertion tolerance. Agents like norepinephrine or dopamine may cause maladaptive hemodynamic responses. Mobilization should follow cardiovascular assessment and be conducted gradually under close supervision [44,45,46].

#### 3.10.3. Surgical Revascularization (CABG/OPCAB) and Postoperative Recovery

Median sternotomy restricts upper-body motion due to wound risks and pain. Postoperative pulmonary complications and cognitive disturbances further hinder early rehab. Emphasis should be placed on lower-limb activity, respiratory therapy, and staged progression [47,48].

#### 3.10.4. Mechanical Complications of Myocardial Infarction

Complications like septal rupture or papillary muscle failure necessitate surgery and prolonged support. Rehabilitation is deferred until cardiorespiratory stability is achieved, with a cautious approach accounting for physical and psychological vulnerability [49,50].

#### 3.10.5. Mechanical Circulatory Support (ECMO, IABP, IMPELLA)

MCS devices require sedation, anticoagulation, and limit movement. Rehab is restricted to passive or assisted techniques; active mobilization under ECMO remains rare and resource-intensive [51,52].

While early rehabilitation post-ROSC is essential, it must be adapted to clinical complexity. Individualized strategies, guided by interdisciplinary teams, ensure safe and context-appropriate recovery pathways.

### 3.11. Neurocognitive Therapy After Hypoxia Following ROSC

Cerebral hypoxia is a primary determinant of long-term neurological outcome after SCA. Even with ROSC, hypoxic–ischemic encephalopathy often leads to persistent cognitive impairments—ranging from fatigue and attentional deficits to profound executive and memory dysfunction. In this context, neurocognitive therapy is a core element of rehabilitation, essential for autonomy and reintegration. The 2022 SCARF study identified fatigue, psychosocial instability, and planning deficits as common yet underrecognized sequelae. Structured assessments revealed impairments in reasoning, processing speed, and emotional regulation, prompting calls for early, personalized interventions integrating cognitive retraining, goal management, and metacognitive strategies within daily activities [13].

Kristensen et al. observed cognitive–motor dissociation: patients with intact motor function post-ICU still demonstrated inefficiencies in task execution weeks later, supporting a dual-channel rehabilitation model combining physical and cognitive therapy [34].

Cognitive profiling with tools such as MoCA or ACE-III initiates the process, but effective recovery requires functional application. Giuvară et al. emphasized functional cognitive therapy (FCT), embedding executive retraining into everyday tasks, paired with guided reflection—yielding improvements in fluency and self-efficacy [39].

Neuropsychiatric symptoms—apathy, disinhibition, depression—further compound deficits. As noted by other authors, these require CBT and, when necessary, pharmacotherapy for mood and anxiety regulation [53].

Digital innovations, as demonstrated in the ROCK trial, offer scalable cognitive monitoring and home-based training. These platforms also support caregiver engagement, reinforcing therapy goals and tracking progress [29].

Neurocognitive rehabilitation post-ROSC is thus restorative, not merely compensatory. It leverages neuroplasticity and directly supports reintegration. Excluding it from standard care risks leaving critical impairments unaddressed, compromising functional recovery and quality of life. Its systematic inclusion in multidisciplinary rehabilitation is evidence-based and essential [35].

### 3.12. Limitations of Neurocognitive Assessment Tools in Post-ROSC Patients’ Recovery

Neurocognitive assessment post-ROSC remains challenging, especially in comatose or severely impaired patients [54]. Widely used tools like the Cerebral Performance Category (CPC) and modified Rankin Scale (mRS) lack sensitivity to subtle deficits and are inadequate for early prognostication in ICU settings [55].

As Rajaje et al. note, most cognitive scales lack validation for post-anoxic encephalopathy, limiting reliability and predictive accuracy [56]. Similarly, EEG and biomarker-based methods, though promising, suffer from variability and lack standardization [57].

These constraints underscore the need for multimodal, longitudinal assessment strategies capable of capturing the evolving neurocognitive recovery in post-cardiac arrest populations.

### 3.13. Individualization of Physiotherapy Protocols After ROSC

Due to the clinical heterogeneity of cardiac arrest survivors, rehabilitation must move beyond standardized physiotherapy to fully individualized protocols. These must reflect each patient’s physiological status, neurological profile, pre-arrest function, and personal goals [2].

Traditional stepwise models—passive mobilization to upright loading—lack flexibility. Nolan et al. proposed a precision approach based on five pillars: neurologic trajectory, cardiopulmonary reserve, consciousness level, fatigue threshold, and physiological feedback [2].

In the ROCK trial, early functional assessments (e.g., sit-to-stand, muscle testing) stratified patients into acuity-based pathways [29].

Kristensen et al. showed that even patients with similar GCS scores vary in task performance, reinforcing the need for task-specific reassessment and real-time adjustment by rehabilitation teams [34].

Some other invastigators highlighted real-time physiological monitoring as key to safe titration of activity intensity. One patient may tolerate tilt-table sessions; another may require only passive therapy due to autonomic instability [53].

Cognitive and emotional status must also guide therapy design. In SCARF, Joshi et al. found that fatigued or emotionally dysregulated patients benefited from short, dual-task interventions, coordinated with neuropsychologists to match cognitive capacity [13].

Individualized physiotherapy is thus essential, not optional. It requires dynamic, interdisciplinary planning to support neuroplasticity, regain independence, and improve life quality after ROSC [38].

### 3.14. The Role of Physiotherapy in the Prevention of Pulmonary and Orthopedic Complications After ROSC

Post-ROSC patients, especially after prolonged ICU stays, face elevated risks of secondary complications unrelated to the initial event. Common issues include pneumonia, atelectasis, ventilator-associated injuries, joint contractures, muscle atrophy, and stiffness [58].

Physiotherapy is essential in mitigating these risks and supporting systemic recovery during both subacute and later stages [59].

Targeted interventions restore respiratory function, aid secretion clearance, enable mobilization, and maintain joint mobility and muscle strength—preventing long-term functional decline [60].

### 3.15. Pulmonary Complications and Respiratory Physiotherapy

Pulmonary complications are common in post-cardiac arrest patients due to immobility, sedation, and mechanical ventilation. Data from the 43rd International Symposium on Intensive Care (2024) indicate that early, individualized respiratory physiotherapy significantly reduces post-extubation pneumonia and oxygenation failure [61].

Interventions such as incentive spirometry, manual chest physiotherapy, PEP therapy, and assisted coughing promote mucociliary clearance and lung re-expansion [62,63].

When hemodynamically stable, early mobilization improves ventilation–perfusion matching and lowers basal atelectasis risk in the first 1–2 weeks post-resuscitation [64].

A 2021 multicenter trial reported a 26% reduction in reintubation and shorter ICU stays when respiratory physiotherapy began within 24 h of ventilator weaning [2].

These findings align with Nolan et al.’s 2021 guidelines, which endorse daily chest physiotherapy in ventilated patients [2,61,65].

### 3.16. Orthopedic Complications and Mobilization Therapy

Musculoskeletal complications—including disuse osteoporosis, joint contractures, myopathy, and joint instability—develop rapidly in immobilized, critically ill patients. If unaddressed, they significantly delay rehabilitation, especially for those with prolonged unconsciousness or severe hypoxic injury. Physiotherapists employ progressive loading—from passive range-of-motion to active-assisted and resistance exercises—to prevent structural decline. At the 44th International Symposium on Intensive Care (2025), early passive mobilization (by ICU day 2) prevented contractures in over 80% of high-risk patients. Neuromuscular electrical stimulation preserved quadriceps mass and supported earlier mobilization. Muscle loss in postural and anti-gravity groups is mitigated by early tilt-table use, in-bed cycling, and gradual verticalization. Kristensen et al. reported that inconsistent joint mobilization led to delayed recovery in 60% of cardiac arrest survivors at discharge [34,66].

### 3.17. Model for Implementing an Interdisciplinary Rehabilitation Approach After ROSC Across Variations in Healthcare Systems

Post-SCA rehabilitation with ROSC requires an interdisciplinary model spanning acute care, convalescence, and long-term follow-up. In the acute phase, ICU, neurology, cardiology, and nursing teams initiate neuroprotection and conduct early prognostic evaluations [2]. Identifying recovery potential enables timely referral to further rehabilitation [67].

During convalescence, typically in rehab or neurology units, individualized interventions are delivered by physiotherapists, speech therapists, psychologists, and occupational therapists, using validated cognitive and functional tools [68].

Structured models in the U.S. and Netherlands exemplify coordinated post-arrest teams offering cognitive–behavioral care [69].

The chronic phase involves outpatient monitoring by GPs, neurologists, and health psychologists, focusing on cognition, quality of life, and community reintegration. Social and long-term care support remain vital [70].

Successful implementation depends on evidence-based protocols, team training, and local adaptation—accounting for resource availability, access to rehabilitation, and referral pathways [2]. Figure 3 shows the post-ROSC algorithm of care including cardiological diagnosis, neurological diagnosis and sequenced rehabilitation.

### 3.18. Timing of Cardiac Interventions in Neurologically Uncertain Post-ROSC Patients: Implications for Rehabilitation and Prognosis

In patients with uncertain neurological status post-ROSC, timing of cardiac interventions—such as PCI or CABG—critically influences recovery potential [71].

Delayed revascularization increases myocardial injury, while premature intervention in comatose patients risks futility if neurological outcomes are poor [56].

Current evidence supports early angiography in suspected acute coronary syndrome, regardless of consciousness, especially with ECG changes or shock [72].

Timely revascularization can preserve function, reduce vasopressor needs, and facilitate earlier rehabilitation—even in unconscious patients [73].

However, neurological prognostication remains unreliable within 72 h, often delaying rehabilitation planning [74].

Multidisciplinary decisions and delayed care withdrawal improve patient selection for combined cardiac and neurorehabilitation [75].

### 3.19. Organ Donation Following ROSC: Management and Clinical Outcomes

ROSC patients who progress to brain death are increasingly important organ donors amid global shortages [76].

Donation after brain death (DBD) post-ROSC is standard in systems with established post-arrest and procurement protocols [77,78].

Management focuses on hemodynamic stability, oxygenation, hormonal therapy (e.g., vasopressin, steroids, levothyroxine), and protection of transplantable organs [79].

Targeted temperature management is withdrawn after brain death, with efforts refocused on perfusion and ischemia minimization [80].

Outcomes for kidneys, livers, and hearts from ROSC donors are comparable to standard DBD donors when ischemic times are controlled and management is protocolized. Slightly higher rates of delayed graft function in renal transplants have been reported, but long-term survival remains equivalent [80,81].

Early transplant team involvement ensures neurologic evaluation, legal compliance, and timely donor preservation [82].

These cases also raise ethical concerns around care continuation without neurologic recovery, reinforcing the need for standardized brain death determination and family-centered decisions [83].
Figure 3Structured algorithm of post-ROSC care pathway: cardiac, neurological, and rehabilitation sequence.
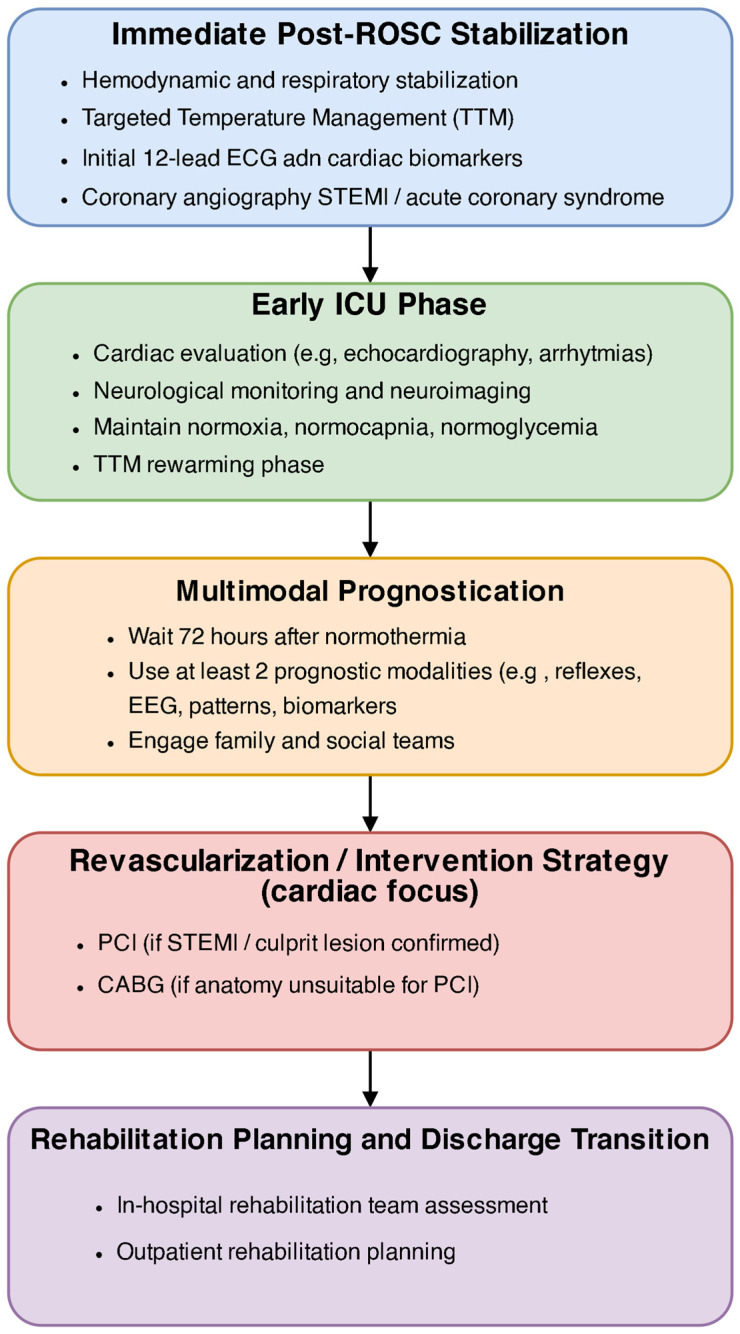


### 3.20. Clinical Indications, Timing, and Reversal Strategies in Targeted Temperature Management (TTM)

Targeted Temperature Management (TTM) remains a core neuroprotective intervention after ROSC in comatose OHCA survivors, especially following shockable rhythms [2].

Current guidelines recommend TTM in all unresponsive patients post-ROSC, with the strongest evidence for ventricular fibrillation or pulseless ventricular tachycardia [84].

TTM should begin within 6 h to reduce reperfusion injury and cerebral metabolism [85].

While earlier protocols targeted 32–34 °C, current recommendations allow 33–36 °C based on individual risk factors (e.g., bleeding, instability) [86].

TTM is maintained for 24 h, followed by controlled rewarming at 0.25–0.5 °C/h to avoid rebound intracranial hypertension and electrolyte shifts [87].

Slow rewarming is especially important in patients with cerebral edema or raised intracranial pressure [88]. Rewarming requires close monitoring for hypokalemia, arrhythmias, and hypotension [89].

Modern protocols utilize automated cooling systems, continuous EEG, and neuromonitoring to detect seizures and support prognostication timing [90].

However, trials like TTM2 have not shown hypothermia to be superior to normothermia, prompting ongoing debate and underscoring the need for individualized application [91].

## 4. Discussion

### 4.1. Contrasting Perspectives on Targeted Temperature Management (TTM) Post-ROSC

TTM has long been central to neuroprotection post-cardiac arrest, with proponents citing improved outcomes in comatose ROSC patients. Tajlil et al. found that, in real-world settings, TTM was linked to increased five-year survival—particularly in patients with prolonged time to ROSC—supporting its use in select populations [92].

However, recent studies challenge its routine use. O’Keefe et al. observed declining TTM utilization without worsened outcomes, aligning with neutral results from TTM2 and questioning the universal benefit of hypothermia across heterogeneous arrest cohorts [93].

### 4.2. Interdisciplinary Integration and Monitoring

Preventing secondary complications through physiotherapy requires close coordination with intensivists, pulmonologists, and neurologists to monitor for contraindications such as intracranial hypertension, arrhythmias, or ventilator dependence [2].

Christensen et al. highlight the need for real-time monitoring of oxygen saturation, hemodynamics, and posture-linked respiratory data to tailor physiotherapy intensity [29].

Pain control, sedation, and patient responsiveness further inform session timing and progression. Occupational therapist involvement ensures alignment with functional goals, incorporating early ADLs—such as transfers, grooming, and feeding—into structured rehabilitation plans [65].

### 4.3. Inequalities in Access to Post-Cardiac Arrest Rehabilitation Across Europe

Despite progress in standardizing resuscitation and post-ROSC care across Europe, access to rehabilitation services remains highly inconsistent. Structural differences in healthcare systems, funding, workforce, and policy prioritization contribute to disparities in recovery outcomes and quality of life [2].

Gräsner et al., in a 28-country analysis, reported wide variation in access to inpatient neurorehabilitation, outpatient physiotherapy, cognitive therapy, and follow-up. While early referral is routine in Germany, the Netherlands, and Nordic countries, it remains delayed or absent in much of Eastern and Southern Europe. Only 38% of centers offer standardized post-arrest rehabilitation pathways [3].

These disparities have clinical consequences. ROSC Outcome Registry data (2022) show that in countries lacking formal rehab programs, return-to-work rates, cognitive outcomes, and quality of life are lower. Survivors without early multidisciplinary care face increased disability, dependency, and post-intensive care syndrome [94,95].

A major barrier is the absence of unified referral criteria. In many regions, ICU teams independently decide rehabilitation timing without structured discharge planning. Nolan et al. advocate integrating rehab into ICU workflows, yet Christensen et al. report that only 21% of ICUs have dedicated rehabilitation coordinators [2,13,29].

Socioeconomic barriers further limit access, especially in rural and underfunded systems. Joshi et al. found that patients in low-income areas often lack basic post-discharge support, information, or continuity of care. Initiatives like the European Cardiac Arrest Rehabilitation Framework (E-CARF) aim to standardize care through interdisciplinary models, but implementation remains uneven. Many countries lack national registries to track outcomes or system gaps [13].

In conclusion, unequal access to rehabilitation is a major, modifiable determinant of survivor outcomes in Europe. Addressing this will require policy alignment, infrastructure investment, and full integration of rehabilitation into national care pathways. Without systemic reform, acute care advances will fail to yield equitable long-term recovery [94].

### 4.4. Variability in the Implementation of Integrated Care Pathways After Cardiac Arrest

Recovery after ROSC is a multi-phase continuum requiring coordinated transitions across ICU, rehabilitation, primary care, and community services. Integrated care pathways (ICPs) were designed to standardize this process, yet implementation remains inconsistent across regions and systems. In the ROCK trial, Christensen et al. evaluated a structured pathway involving early rehabilitation assessment, cognitive screening, scheduled follow-up, and community support. While associated with improved return-to-work and functional independence, outcomes varied by site. Hospitals with dedicated coordinators and registries showed higher adherence; others experienced fragmented transitions and reduced engagement [29].

Key barriers included lack of discharge protocols, limited neurorehabilitation access, unclear provider roles, and poor communication with primary care—leading to missed referrals and caregiver exclusion. Gräsner et al. similarly reported major cross-country variation in care bundle adoption. High-resource settings integrated ICPs into electronic records; lower-resource areas relied on clinician discretion, resulting in uneven application [3].

Christensen et al. noted that successful ICPs depend on interdisciplinary ownership, local adaptation, structured handovers, and regular team meetings—ensuring consistency despite system variability [40].

Joshi et al. found that the absence of structured pathways led to care mismatches, non-referral, and post-discharge decline by three months [13].

Nolan et al. emphasized that pathway presence alone is insufficient; effectiveness depends on institutional leadership, monitoring systems, and prioritization of neurocognitive and caregiver support [2].

In sum, ICPs are essential to optimize post-arrest recovery, but require standardization, digital integration, staff training, and systemic coordination. Without such reforms, benefits seen in trials like ROCK will remain confined to select settings [29].

### 4.5. The Need for Randomized Controlled Trials Evaluating Rehabilitation Protocols After ROSC

Despite growing consensus on the importance of rehabilitation after ROSC, the supporting evidence remains limited. Current ERC and AHA guidelines (2021) draw primarily on expert opinion, retrospective analyses, and small-scale feasibility studies rather than robust randomized controlled trials (RCTs) [6].

This deficit reflects the heterogeneity of the post-ROSC population, in whom variable neurological profiles complicate standardization. While acute-phase interventions—such as TTM and hemodynamic support—have been rigorously studied, rehabilitation remains under-evaluated. The SCARF study by Joshi et al. confirmed feasibility and functional gains with structured rehabilitation but lacked randomization and statistical power, prompting calls for large-scale trials [13].

Similarly, the pragmatic ROCK trial by Christensen et al. demonstrated improved outcomes with an interdisciplinary pathway but suffered from unblinded assessments and site variability, underscoring the need for standardized protocols and stratified designs [29].

Overall, there is a clear need for large-scale, rigorously designed randomized controlled trials to establish the efficacy, timing, and optimal components of rehabilitation for cardiac arrest survivors. Without such evidence, practice will continue to rely on extrapolation and local adaptation, limiting the potential for consistent, evidence-based care.

The need for randomized controlled trials extends beyond general rehabilitation efficacy to critical components of intervention design, including:optimal timing of initiation, such as comparing outcomes of rehabilitation started within 24 h post-ROSC versus after ICU dischargedetermination of ideal intensity and frequency for mobilization protocols to prevent deconditioning while avoiding physiological stresscomparative effectiveness of home-based rehabilitation models versus structured inpatient programs in terms of adherence, function, and reintegrationvalidation of neurocognitive therapy algorithms stratified by baseline cognitive profiles using tools like the Montreal Cognitive Assessment or EEG patternsassessment of virtual and digital cognitive–motor rehabilitation platforms in improving cognitive outcomes and access to therapy, particularly in patients with limited mobility or rural access barriers.

A 2023 systematic review by Joshi et al. found most existing trials to be at high risk of bias, lacking standardized outcome measures such as EQ-5D or WHODAS 2.0, and failing to control for key confounders [13].

Funding priorities further hinder progress. As contributing scientists noted in theirs studies, critical care research remains biased toward acute-phase interventions, while post-ICU survivorship receives minimal support [61].

Initiatives such as EU-CARREST and E-CARF have proposed harmonized research frameworks and urged increased investment in post-ROSC rehabilitation trials, analogous to the impact of TTM and COACT on acute care [38].

Beyond methodological urgency lies an ethical imperative. With 50–70% of ROSC survivors experiencing lasting deficits, continued neglect of structured rehabilitation—based solely on evidentiary limitations—risks perpetuating preventable disability. Carefully designed trials with adaptive methods and stakeholder engagement would align scientific rigor with ethical responsibility [38].

In conclusion, without high-certainty data from RCTs, rehabilitation after cardiac arrest remains inconsistently applied and insufficiently optimized. Bridging this gap is essential to move beyond survival toward meaningful, evidence-based recovery [38].

### 4.6. Developing Pathways for Return to Work and Social Reintegration After Cardiac Arrest

Survival after cardiac arrest initiates a prolonged recovery process in which physical and cognitive rehabilitation must be accompanied by structured return-to-work (RTW) and social reintegration planning. These higher-order outcomes are essential to autonomy, identity, and psychological health, yet remain underrepresented in clinical frameworks and policy. The SCARF study by Joshi et al. revealed that despite regained basic function, many survivors face persistent fatigue, memory deficits, anxiety, and impaired executive function, all of which impede RTW. Structured vocational support and CBT markedly improved re-employment rates within six months [13].

Similarly, the ROCK trial reported a 64% RTW rate at one year in centers with integrated support, versus 37% without [29].

Kristensen et al. emphasized that subtle cognitive–motor impairments—undetectable in standard exams—adversely affect occupational and social functioning. Tools such as the Work and Social Adjustment Scale, WHODAS 2.0, and the RTW Self-Efficacy Questionnaire are recommended for individualized readiness assessment [34].

Psychosocial rehabilitation is central. As Mion and Keeble argue, self-confidence, emotional stability, and restored identity are as critical as physical recovery. Interventions including peer mentoring, vocational coaching, and role-focused therapy support successful reintegration [36].

At the systemic level, the 2023 ESC position paper called for standardized RTW protocols across EU states, integrating early occupational health input, financial counseling, and cognitive-job matching [38].

Supporting this, a 2023 Austena study showed that early employer engagement doubled RTW likelihood, particularly when accommodations were provided (e.g., flexible hours, cognitive task modification) [96].

Social reintegration extends beyond employment. Driving, family interaction, and community participation are often impaired. Giuvară et al. reported that nearly half of survivors exhibit social withdrawal without targeted support—underscoring the value of life-skills training and structured family reintegration [39].

In conclusion, RTW and social re-engagement are not ancillary but foundational rehabilitation goals. Embedding them into structured, interdisciplinary care ensures that recovery is measured not only by survival, but by restoration of agency, identity, and full life participation [97].

### 4.7. Innovations in Brain Function Assessment

Timely and precise evaluation of cerebral function following cardiac arrest remains fundamental for prognostication, therapeutic planning, and rehabilitation stratification. Although conventional modalities—such as EEG, CT imaging, and neurological examination—constitute the clinical foundation, the advent of neuronal and glial injury biomarkers has introduced a novel paradigm in the early detection of post-hypoxic encephalopathy. These markers provide objective quantification of brain injury severity in the acute phase after ROSC. Wihersaari et al. demonstrated that elevated serum concentrations of neurofilament light chain (NfL) and glial fibrillary acidic protein (GFAP) within the first 72 h post-resuscitation strongly correlate with subsequent functional and cognitive deficits. Particularly, elevated NfL at 24 and 48 h predicted sustained executive dysfunction and diminished return-to-work rates, even in clinically stable patients—underscoring the limitations of traditional clinical assessments and the diagnostic value of subclinical biomarkers [98].

Further investigations by Klitholm et al. and Wihersaari et al. validated the prognostic superiority of NfL and tau over conventional modalities such as somatosensory evoked potentials. NfL levels demonstrated a predictable rise within 24 h of insult, peaking by 48–72 h, thereby offering an early, quantifiable index of neuronal injury [99,100,101].

Additionally, UCH-L1 has emerged as a candidate marker capable of differentiating cortical from subcortical damage. Amoo et al. (2023) emphasized that UCH-L1 stratification may inform rehabilitation pacing and intensity in early recovery phases [98].

When incorporated into multimodal prognostic frameworks–as seen in TTM-2 and COMACARE—these biomarkers significantly enhance predictive accuracy. Patients with low biomarker levels but adverse EEG profiles occasionally exhibited favorable outcomes, suggesting that biomarker-informed protocols could mitigate premature care withdrawal and ensure appropriate rehabilitation targeting. Advances in bedside diagnostics have facilitated the integration of point-of-care biomarker testing; a 2024 study published in Critical Care Medicine demonstrated rapid electrochemiluminescence-based NfL assays, yielding results within 90 min and supporting real-time ICU deployment [35].

However, biomarker interpretation requires contextualization. Age, neurodegenerative comorbidities, and systemic illness can influence baseline values. As emphasized by Nolan et al. and Wihersaari et al., longitudinal measurements and individualized thresholds are essential for diagnostic accuracy. Moreover, technological disparities limit access to rapid testing in resource-constrained settings [2,34].

Despite these challenges, biomarker-guided evaluation represents a critical advancement in post-cardiac arrest care. By enabling early, individualized classification of brain injury, these tools strengthen prognostic precision, support tailored rehabilitation planning, and inform ethically sound decisions regarding care continuation [56,102].

### 4.8. Gaps of Knowledge

There remains no unified standard regarding when and how intensively to initiate neurorehabilitation after cardiac arrest, resulting in variation across clinical settings [103].

Although preliminary evidence supports early intervention, high-quality randomized trials confirming its long-term benefits remain absent [104].

Furthermore, debate persists over which cognitive domains—particularly executive and affective functions—should be prioritized, as current protocols often overlook these areas [105].

Research is further limited by inconsistent outcome metrics and the lack of standardized rehabilitation frameworks [106].

The role of comorbidities, such as dementia or frailty, in modulating rehabilitation outcomes is poorly understood [46].

Finally, inequities in access to specialized services and structured follow-up disproportionately affect patients in low-resource and rural regions [107].

### 4.9. Limitations of Study

This narrative review is subject to several methodological limitations. Most notably, the absence of quantitative meta-analysis limits generalizability and precision in estimating treatment effects. Selection bias may also be present, as the review focused on peer-reviewed sources and recent guidelines, potentially favoring studies with positive outcomes. Significant heterogeneity across included studies—regarding patient cohorts, intervention protocols, and outcome measures—further impedes direct comparison. Exclusion of gray literature and unpublished data may have overlooked relevant findings. Moreover, disparities in regional healthcare infrastructure constrain the universal applicability of proposed interdisciplinary models. Critically, the lack of high-quality randomized trials addressing the timing, intensity, and structure of post-ROSC rehabilitation weakens the current evidence base. Future research should prioritize large-scale controlled trials to support guideline development and clinical standardization. Table 2 provides a precise summary of studies and early rehabilitation and its outcomes after ROSC.

### 4.10. Future Research

Future investigations must clarify the optimal timing, intensity, and duration of multidisciplinary rehabilitation following ROSC to enhance recovery and reduce disability [75]. Large randomized controlled trials are urgently needed to assess standardized neurorehabilitation protocols and their impact on both survival and quality of life [108]. Development of sensitive, domain-specific cognitive assessment tools tailored to post-anoxic injury is also a key research priority [109]. Furthermore, the influence of pre-morbid conditions—such as frailty, dementia, and psychiatric disorders—on rehabilitation outcomes remains poorly understood and warrants further study [110]. International consensus on core outcome measures is essential to enable comparability and meta-analysis across trials [51]. Lastly, equity-focused research should address disparities in rehabilitation access, particularly among rural and underserved populations [111].

## 5. Conclusions

Return of spontaneous circulation following sudden cardiac arrest initiates a prolonged and multifaceted recovery that demands continuous interdisciplinary involvement. While advances in resuscitation have improved survival rates, attention must now prioritize post-discharge outcomes. Rehabilitation plays a critical role in shaping neurological prognosis, functional autonomy, and reintegration into society.Evidence across all stages of recovery—from ICU to community-based follow-up—consistently supports individualized, structured, and multidisciplinary rehabilitation as foundational. Through targeted interventions, rehabilitation enables restoration of motor skills, cognitive function, emotional regulation, and daily participation.Optimal recovery requires coordinated efforts among cardiologists, neurologists, physiotherapists, occupational therapists, and neuropsychologists, integrated within a unified care framework. Family involvement is equally vital; informed and supported caregivers are central to ensuring adherence and long-term continuity of care.Significant disparities in rehabilitation access persist. Variability in service provision and the absence of standardized pathways undermine recovery potential. These inequities necessitate systemic reforms, including the implementation of care-integrated protocols and equitable investment in rehabilitation resources.Recovery after cardiac arrest should be redefined as an active process of reestablishing physical, cognitive, vocational, and social identity. Rehabilitation must be viewed not as ancillary, but as an indispensable extension of resuscitation. A coordinated, patient-centered approach is essential to translating survival into a meaningful return to life.

## Figures and Tables

**Figure 1 healthcare-13-01865-f001:**
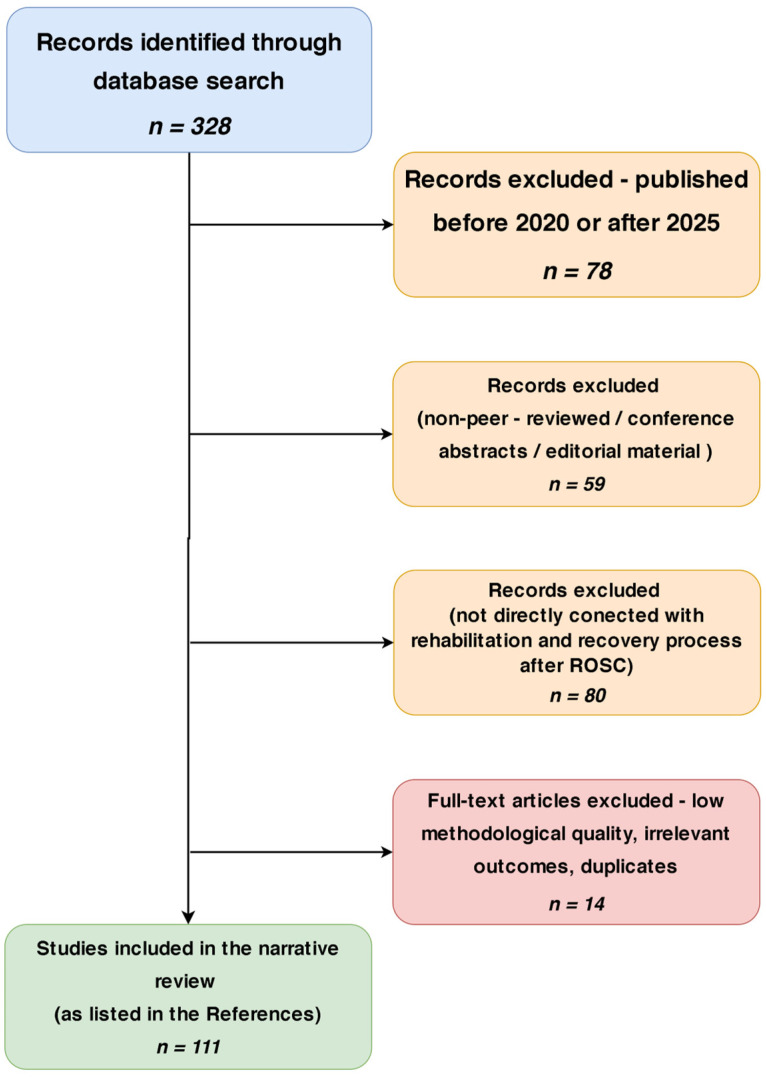
Schematic study selection process for the narrative review.

**Figure 2 healthcare-13-01865-f002:**
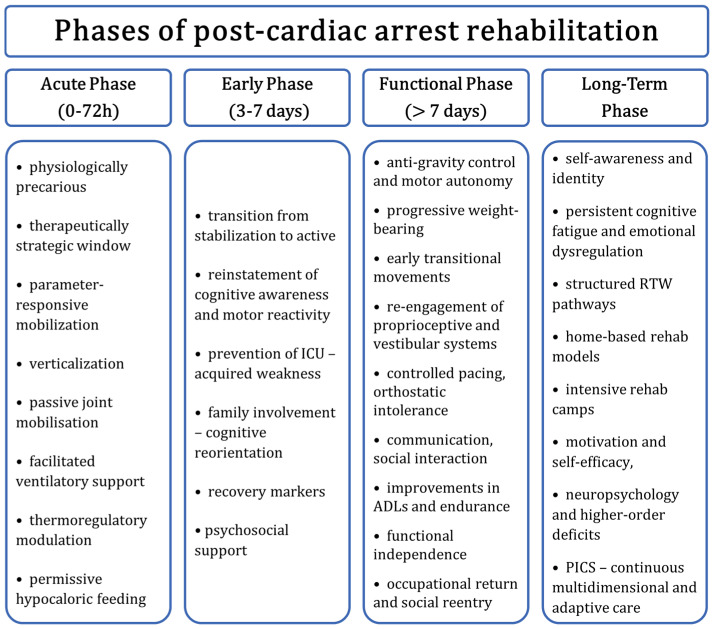
Proposed progression of rehabilitation following ROSC after SCA, including acute care in the ICU (stabilization, neuroprotection, possible targeted temperature management), subacute rehabilitation (early physical and cognitive recovery), and long-term outpatient follow-up (community-based reintegration, neuropsychological therapy). Abbreviations: ROSC—return of spontaneous circulation; SCA—sudden cardiac arrest; ICU—intensive care unit; TTM—targeted temperature management; CPC—cerebral performance category.

**Table 1 healthcare-13-01865-t001:** Inclusion and exclusion criteria for articles considered in this narrative review.

Inclusion Criteria	Exclusion Criteria
Peer-reviewed articles published between 2020 and 2025	Articles published prior to 2020 or after 2025
Publications subjected to editorial and peer review processes	Non-peer-reviewed works, editorials, letters to the editor, or conference abstracts lacking full-text availability
Studies explicitly addressing rehabilitation following return of spontaneous circulation (ROSC) after SCA	Studies not related to post-ROSC rehabilitation or focusing exclusively on acute or resuscitative care
Articles aligned with evidence-based official and newest clinical guidelines.	Articles omitting or disregarding contemporary resuscitation or rehabilitation guidelines
Original research: randomized controlled trials, cohort studies, observational analyses, and structured reviews	Publications with low methodological quality, lacking a clearly defined study design or analytical rigor
Studies addressing neurological, cardiopulmonary, neurocognitive, or psychosocial rehabilitation domains	Articles solely focusing on preclinical, pharmacological, or procedural resuscitation measures
Articles published in the English language	Publications in languages other than English
Full-text availability for critical appraisal and synthesis	Articles without accessible full text, with limited eligibility for methodological evaluation

**Table 2 healthcare-13-01865-t002:** Summary of studies on rehabilitation and outcomes after ROSC. This table summarizes clinical and experimental studies, focusing on neurological outcomes, prognostic tools, and therapeutic interventions. Abbreviations: ROSC—return of spontaneous circulation, TTM—targeted temperature management, EEG—electroencephalography, CPC—cerebral performance category, mRS—modified Rankin scale, GCS—Glasgow coma scale, SEP—somatosensory evoked potentials, RTW—return to work, OHCA—out-of-hospital cardiac arrest, ECPR—extracorporeal cardiopulmonary resuscitation, ICU—intensive care unit, ERC—European Resuscitation Council, MoCA—Montreal Cognitive Assessment, HADS—hospital anxiety and depression scale.

No.	Author (Year)	Population (N)	Intervention	Outcome	CPC/mRS	Rehabilitation Stage	Notes
1	Harding L et al. (2024) [99]	20 (animal model)	SEP measurement after ROSC	Lower SEP amplitudes correlated with more severe brain injury.	Experimental model—no CPC assessment.	Early post-ROSC phase (0–72 h)	Indicates the utility of SEP as a prognostic tool.
2	Bakhsh A et al. (2024) [17]	86 in-hospital cardiac arrest patients	Fever prevention vs. no temperature control.	CPC 1–2: 52% vs. 31% (*p* = 0.03)	CPC	Acute phase of intensive care	Statistically significant difference —better neurological outcomes.
3	Sepúlveda P et al. (2025) [62]	ICU patients (narrative review)	Protocolized early mobilization (screening, PT-led activity within ~72 h)	Reduced duration of mechanical ventilation and ICU stay	Not assessed	Early mobilization phase	Emphasizes multidisciplinary teamwork, education, safety screening (hemodynamic, respiratory, neurological), func-tional mobilization and outcome documentation
4	Shen X et al. (2023) [35]	93 coma patients after cardiac arrest	EEG analysis combined with a dynamic risk model.	EEG variability correlated with CPC 1–2 at 6 months.	CPC	0–7 days (prognostic phase)	Potential for individualized rehabilitation based on EEG findings.
5	Nolan JP et al. (2021) [2]	Pooled data from RCTs and registries	ERC guidelines: rehabilitation, neuro-observation, TTM.	CPC 1–2 in 48–55% of patients after OHCA.	CPC	In-hospital and post-discharge	Strong recommendation for a multidisciplinary approach.
6	Gräsner J-T et al. (2021) [3]	EU registry: >250,000 OHCA cases	Observation: contribution of rehabilitation to survival.	ROSC: 27%; Survival to discharge: 9.4%	No data available, mortality only.	No intervention, population-based analysis.	Data support the need for post-ROSC care.
7	Christensen J et al. (2024) [29]	Planned: 300 OHCA survivors	Multidisciplinary rehabilitation focused on RTW.	Study protocol —results unpublished.	Planned: mRS, RTW.	Post-discharge (long-term RTW)	To be utilized upon the publication of results.
8	Wang J et al. (2024) [18]	786 patients from 12 studies (ECPR + TTM).	Comparison of TTM: 32 °C, 33 °C, 36 °C, no temperature control.	TTM at 33 °C: CPC 1–2 in 56% vs. no TTM: 42%.	CPC	Intensive care phase (24–72 h)	Best outcomes observed at 33 °C; meta-analysis.
9	Joshi VL et al. (2022) [13]	82 post-cardiac arrest survivors referred to residential rehab	Structured residential rehabilitation targeting fatigue, cognition, and psychological sequelae	88% program completion; significant reduction in fatigue (mean improvement: 13.5 points, *p* < 0.001); improved MoCA and HADS scores	MoCA, HADS; no CPC	Post-discharge (residential, subacute phase)	Demonstrated feasibility and clinical benefit of residential multimodal rehabilitation; supports implementation in recovery pathways.
10	Giuvară S et al. (2024) [39]	1 female OHCA patient with comorbidities	Physiotherapy, mobilization, and dietary education.	After 4 weeks: improved limb strength, increased independence (Barthel + 20%).	No numerical CPC data available.	Post-hospital rehabilitation	Single case—clinical illustration.

## Data Availability

Not applicable.

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
