# Peer review of "Current Perspectives on Rehabilitation Following Return of Spontaneous Circulation After Sudden Cardiac Arrest: A Narrative Review"

_healthcare, 2025, doi:10.3390/healthcare13151865_

Round 1
Reviewer 1 Report
Comments and Suggestions for Authors
This narrative review tackles a critically important and often underrepresented aspect of cardiac care—the multidisciplinary rehabilitation process following return of spontaneous circulation (ROSC) after sudden cardiac arrest (SCA). The topic is both timely and relevant, particularly in light of growing recognition that survival from cardiac arrest does not equate to full recovery without structured and sustained rehabilitation. The manuscript does an excellent job of framing rehabilitation not as a supplementary phase but as an essential pillar of care for survivors. The structured format, organized around acute, subacute, functional, and long-term rehabilitation phases, enhances the review’s utility for both clinical and academic audiences. Additionally, the authors’ emphasis on cognitive, psychosocial, and neurorehabilitation domains, often overlooked in cardiac literature; adds significant value and depth.
The manuscript demonstrates commendable breadth in scope. It comprehensively synthesizes recent studies, guidelines, and expert opinions, drawing from a range of high-quality sources, including the AHA, ERC, and ILCOR. It clearly outlines the pathophysiological underpinnings of post-cardiac arrest syndrome (PCAS) and effectively bridges those mechanisms to the rationale for early and tailored rehabilitation. The narrative is rich in clinical relevance, integrating real-world barriers such as healthcare access disparities, staffing limitations, and patient adherence issues. Moreover, the inclusion of neurocognitive therapy and family support as integral aspects of recovery is both progressive and grounded in current evidence.
That said, the manuscript could benefit from some refinement. First, while it is well-referenced and logically structured, it is also exceedingly dense. Several sections, particularly those describing study outcomes or physiotherapeutic techniques, are overloaded with detail that could overwhelm readers. A more concise writing style would make the manuscript more digestible without sacrificing depth. Additionally, some repetition exists—certain concepts such as ICU-acquired weakness, the phases of rehabilitation, and the importance of family engagement are reiterated across multiple sections. These could be consolidated or referenced more efficiently to improve flow.
Another area for improvement is the balance between narrative synthesis and critical appraisal. While the manuscript does summarize a wealth of literature, it often does so descriptively rather than analytically. For instance, contrasting perspectives on the utility of targeted temperature management (TTM), or discussing the limitations of existing neurocognitive tools, would enrich the reader’s understanding. Furthermore, the discussion would benefit from clearer identification of knowledge gaps or controversies in the field, helping to shape future research directions.
The figures and tables, particularly Figure 1, serve as useful visual summaries of the rehabilitation process. However, figure legends could be expanded for clarity, and formatting across sections would benefit from minor cleanup (e.g., aligning abbreviations, ensuring citation consistency). Additionally, a final section summarizing practical recommendations or a proposed framework for implementing interdisciplinary ROSC rehabilitation models across different healthcare systems would elevate the impact of the review.
Author Response
Response to Reviewer #1 Comments
We sincerely thank the Reviewer #1 for the time and effort devoted to the evaluation of our manuscript. We highly appreciate the constructive comments and valuable suggestions, which have helped us to improve the quality and clarity of the paper. Below, we provide a point-by-point response to each of the Reviewer’s remarks, indicating the changes made to the manuscript accordingly. All modifications are marked in the revised version for ease of reference.
“First, while it is well-referenced and logically structured, it is also exceedingly dense. Several sections, particularly those describing study outcomes or physiotherapeutic techniques, are overloaded with detail that could overwhelm readers. A more concise writing style would make the manuscript more digestible without sacrificing depth.”
We thank the Reviewer for this valuable observation. In response, we have substantially revised the indicated sections to enhance clarity and readability. Specifically, descriptions of study outcomes and physiotherapeutic techniques have been carefully condensed and streamlined. Excessive elaboration was removed, and a more concise and focused writing style has been adopted throughout the manuscript. As a result, the overall textual content has been reduced by approximately 40%. These changes improve the flow of information and make the manuscript more accessible, without compromising scientific accuracy or depth.
“Additionally, some repetition exists—certain concepts such as ICU-acquired weakness, the phases of rehabilitation, and the importance of family engagement are reiterated across multiple sections. These could be consolidated or referenced more efficiently to improve flow.”
We carefully identified and removed all instances of unnecessary repetition throughout the manuscript. Redundant references to ICU-acquired weakness, the phases of rehabilitation, and the role of family engagement have been eliminated. The relevant content has been consolidated and retained only at its first occurrence in the text. This revision improves textual coherence, enhances the logical flow, and avoids redundancy, while preserving the integrity of the discussed concepts.
“Another area for improvement is the balance between narrative synthesis and critical appraisal. While the manuscript does summarize a wealth of literature, it often does so descriptively rather than analytically. For instance, contrasting perspectives on the utility of targeted temperature management (TTM), or discussing the limitations of existing neurocognitive tools, would enrich the reader’s understanding.”
In response, we have added two dedicated subsections that provide focused critical appraisal of the specified topics. Section 3.12, Limitations of neurocognitive assessment tools in post-ROSC patients recovery, offers a concise yet comprehensive evaluation of methodological and clinical constraints associated with commonly used tools. Additionally, Section 4.1, Contrasting perspectives on Targeted Temperature Management (TTM) Post-ROSC, presents a balanced discussion of the divergent evidence and ongoing debate surrounding TTM efficacy. These additions enhance the manuscript’s analytical depth and contribute to a more nuanced interpretation of the current literature.
“Furthermore, the discussion would benefit from clearer identification of knowledge gaps or controversies in the field, helping to shape future research directions.”
We thank the Reviewer for this important recommendation. In response, we have expanded the discussion section by introducing two new structured subsections: Section 4.8, Gaps of knowledge, which outlines the key unresolved issues and limitations within the current evidence base, and Section 4.10, Future research, which proposes targeted directions for further investigation. These additions are concise and thematically organized to improve clarity and emphasize the interdependence between existing uncertainties and future research priorities.
“The figures and tables, particularly Figure 1, serve as useful visual summaries of the rehabilitation process. However, figure legends could be expanded for clarity, and formatting across sections would benefit from minor cleanup (e.g., aligning abbreviations, ensuring citation consistency). “
We thank the Reviewer for highlighting the value of Figure 1 and for the constructive suggestions. In response, we have refined the content of Figure 1 to improve its clarity and informational value. The accompanying legend has been expanded to provide a more detailed explanation of each component, thereby enhancing the reader’s understanding of the depicted rehabilitation process. Additionally, formatting inconsistencies across the manuscript—including abbreviation alignment and citation formatting—have been systematically reviewed and corrected.
“Additionally, a final section summarizing practical recommendations or a proposed framework for implementing interdisciplinary ROSC rehabilitation models across different healthcare systems would elevate the impact of the review.”
We greatly appreciate this constructive suggestion. In direct response, we have added Section 3.17, Model for implementing an interdisciplinary rehabilitation approach after ROSC across variation of healthcare systems. This subsection outlines a practical, adaptable framework that integrates key elements of post-ROSC rehabilitation into diverse healthcare contexts. The proposed model is based on the synthesized evidence and aims to provide clinicians and policymakers with a structured reference for implementation across settings with differing resources and infrastructures.

Reviewer 2 Report
Comments and Suggestions for Authors
I have carefully read the article titled "Current perspectives on rehabilitation following return of spontaneous circulation after sudden cardiac arrest: a narrative" that was sent to me for evaluation. My comments, criticisms and suggestions are listed below:
In this review, the authors emphasise the importance of structured, multidisciplinary rehabilitation after ROSC and the need for integrated physiotherapy, neurocognitive therapy and psychosocial support. I congratulate the authors for the effort they put into review this important topic.
In the presence of acute coronary syndrome, it would be good to discuss the effects of emergency coronary intervention, hemodynamic management, surgical revascularization, treatment of mechanical complications of myocardial infarction (medical or surgical), and the application of advanced mechanical support systems on rehabilitation processes.
If there are cardiac pathologies that need to be corrected in cases with uncertain neurological status after ROSC, what are the effects of the strategies and timing to be applied on the rehabilitation process and prognosis? This important issue should be discussed in the literature.
It would be appropriate to discuss the specific indications for Targeted Temperature Management (induced hypothermia), the timing and duration of treatment, and subsequent reversal methods with literature review.
Considering the current donor shortage, a significant portion of organ transplantation procedures use donor organs from ROSC patients. Comments on the management strategies and clinical outcomes in this situation should be made.
Explaining the main topics related to post-ROCS care (such as cardiac evaluation, interventional neurological evaluation and treatment, prognostication, in-hospital and post-discharge rehabilitation) with algorithms or schematic visuals will enhance the ease of reading and understanding of the article.
Best Regards.
Author Response
Reviewer 2#
“In the presence of acute coronary syndrome, it would be good to discuss the effects of emergency coronary intervention, hemodynamic management, surgical revascularization, treatment of mechanical complications of myocardial infarction (medical or surgical), and the application of advanced mechanical support systems on rehabilitation processes.”
We thank the Reviewer for this valuable suggestion. In response, we have included Section 3.10, Clinical barriers and complications affecting early rehabilitation after ROSC, which comprehensively addresses the impact of acute coronary syndrome–related interventions on post-resuscitation rehabilitation. This section discusses the effects of emergency coronary intervention, hemodynamic instability, surgical and medical management of myocardial complications, and the role of mechanical circulatory support, all considered from the perspective of physiotherapy planning and execution. The addition enhances the clinical relevance of the manuscript and strengthens its applicability to real-world rehabilitation scenarios.
“If there are cardiac pathologies that need to be corrected in cases with uncertain neurological status after ROSC, what are the effects of the strategies and timing to be applied on the rehabilitation process and prognosis? This important issue should be discussed in the literature.”
We thank the Reviewer for drawing attention to this critical and underexplored topic. In response, we have added Section 3.18, Timing of cardiac interventions in neurologically uncertain Post-ROSC patients: implications for rehabilitation and prognosis. This subsection examines current evidence and clinical considerations regarding the timing and selection of cardiac interventions in patients with unclear neurological outcomes following ROSC. The implications of these strategies for early rehabilitation planning and long-term prognosis are analyzed, thereby expanding the manuscript’s relevance to interdisciplinary care decision-making.
“It would be appropriate to discuss the specific indications for Targeted Temperature Management (induced hypothermia), the timing and duration of treatment, and subsequent reversal methods with literature review.”
We appreciate the Reviewer’s suggestion to elaborate on this clinically significant aspect. In response, we have added Section 3.20, Clinical indications, timing, and reversal strategies in Targeted Temperature Management (TTM). This subsection provides a comprehensive and clearly structured overview of the current evidence regarding patient selection, optimal initiation timing, recommended duration of TTM, and controlled rewarming protocols. The content is based on up-to-date literature and is intended to enhance the clinical applicability and depth of the manuscript.
“Considering the current donor shortage, a significant portion of organ transplantation procedures use donor organs from ROSC patients. Comments on the management strategies and clinical outcomes in this situation should be made.”
We thank the Reviewer for highlighting this increasingly relevant issue. In response, we have introduced Section 3.19, Organ donation following ROSC: Management and clinical outcomes. This subsection offers a clear and structured review of the available literature on post-ROSC organ donation, including clinical criteria, management strategies to optimize organ viability, and reported outcomes. The addition expands the manuscript’s scope and addresses an important intersection between resuscitation medicine and transplant ethics.
“Explaining the main topics related to post-ROCS care (such as cardiac evaluation, interventional neurological evaluation and treatment, prognostication, in-hospital and post-discharge rehabilitation) with algorithms or schematic visuals will enhance the ease of reading and understanding of the article.”
We thank the Reviewer for this valuable suggestion aimed at improving the clarity and educational value of the manuscript. In response, we have created and inserted a new schematic visual, Figure 3, placed immediately before the Discussion section. This figure synthesizes key components of post-ROSC care—including cardiac and neurological evaluation, prognostication, and the rehabilitation continuum—into a single, coherent algorithm. The visual format is intended to facilitate understanding, enhance knowledge translation, and support practical application of the content.

Reviewer 3 Report
Comments and Suggestions for Authors
The authors present an extensive and very well documented narrative review about the rehabilitation following the return of spontaneous circulation after a sudden cardiac arrest. The review includes a sum of the most relevant studies focusing on this aspect published from 2020-2025. Their work emphasis the need for the development of management protocols for these patients, and the multidisciplinary dimension of this first glance major cardiological pathology (neurological recovery, functional independence, and societal reintegration with: occupational therapy, work re-entry, and of course family counseling and education).
The major inconvenience is represented by the length of their work, that makes it hard to follow and summarize the information included. Perhaps, if it is possible, a shortening of the aspects approached, or charts to summarize the most relevant aspects, perhaps doubled with some statistics if available from the included articles, will help improve readability and understanding.
Besides this major aspect some minor issues need to be addressed.
Abstract section: Please explain ”ICU” abbreviation.
Introduction section: Please explain ”CPR” abbreviation.
Materials and Methods: Please explain ”AHA” abbreviation and state the inclusion and exclusion criteria and perhaps make a flow chart of the entire article selection process.
Results: Please make another flow chart or table in which you synthetize the results for a better and easier understanding due to the fact that the article presents in a very exhaustive form the results of the included articles and their statistics results.
Please explain the abbreviations ”IL-6”, ”TNF”, ”SCARF”, ”WHO”, ”ROCK”, ”WHODAS 2.0”, EQ-5D”, JAMA”, ”COACT”, ”EU”, ”CT”, COMACARE”,
Figure 1- acute phase: the 4th line ”passive joint articulation” – please correct, maybe ”passive joint mobilization” .
Row 407: extra space after Joshi et al.

Author Response
Reviewer 3#
“The major inconvenience is represented by the length of their work, that makes it hard to follow and summarize the information included. Perhaps, if it is possible, a shortening of the aspects approached, or charts to summarize the most relevant aspects, perhaps doubled with some statistics if available from the included articles, will help improve readability and understanding.”
We appreciate the Reviewer’s constructive feedback regarding the manuscript's length and density. In response, we have significantly shortened and consolidated the main text, reducing its volume by approximately 40% to enhance clarity and focus. Additionally, we have included a new summary table—Table 2—which presents key statistical data extracted from the reviewed literature in a clear, structured format. This addition is intended to support easier navigation and synthesis of the most relevant findings, thereby improving overall readability and comprehension.
“Besides this major aspect some minor issues need to be addressed.
Abstract section: Please explain ”ICU” abbreviation.
Introduction section: Please explain ”CPR” abbreviation.
Materials and Methods: Please explain ”AHA” abbreviation and state the inclusion and exclusion criteria and perhaps make a flow chart of the entire article selection process.”
We thank the Reviewer for these detailed and helpful observations. All requested corrections have been implemented as follows: the abbreviations “ICU” (Intensive Care Unit), “CPR” (Cardiopulmonary Resuscitation), and “AHA” (American Heart Association) have been defined upon first mention in the respective sections. Moreover, the inclusion and exclusion criteria have been clearly specified in the Materials and Methods section as Tabel 1. In addition, a flowchart illustrating the article selection process has been added as Figure 1, providing a transparent overview of the literature review methodology.
“Results: Please make another flow chart or table in which you synthetize the results for a better and easier understanding due to the fact that the article presents in a very exhaustive form the results of the included articles and their statistics results.”
We thank the Reviewer for this important suggestion aimed at improving the manuscript’s structure and accessibility. In response, we have shortened and consolidated the main text by approximately 40% to improve clarity and eliminate redundancies. Furthermore, we have introduced Table 2, which synthetically presents key statistical findings from the reviewed literature. To further enhance readability and information flow, we have added two new flowcharts—Figure 2 and Figure 3—which summarize the major components and decision pathways related to post-ROSC care. Together, these modifications improve the manuscript’s overall coherence, facilitate data interpretation, and support reader engagement.
“Please explain the abbreviations ”IL-6”, ”TNF”, ”SCARF”, ”WHO”, ”ROCK”, ”WHODAS 2.0”, EQ-5D”, JAMA”, ”COACT”, ”EU”, ”CT”, COMACARE”,”
“Figure 1- acute phase: the 4th line ”passive joint articulation” – please correct, maybe ”passive joint mobilization” .
Row 407: extra space after Joshi et al.”
We thank the Reviewer for the attentive review and helpful editorial suggestions. All abbreviations listed have now been defined upon their first appearance in the manuscript to improve clarity. The phrase “passive joint articulation” in Figure 1 has been corrected to “passive joint mobilization” for terminological accuracy. Additionally, the extra space after “Joshi et al.” in line 407 has been removed.

Round 2
Reviewer 1 Report
Comments and Suggestions for Authors
It can be accepted now.
Author Response
The authors appreciate the suggestions and involvement of the reviewer, which made the article better.
Reviewer 2 Report
Comments and Suggestions for Authors
I carefully read the revised version of the article "Current perspectives on rehabilitation following return of spontaneous circulation after sudden cardiac arrest: a narrative review," which was submitted to me for reconsideration. First of all, I congratulate the authors for the revisions they made based on reviewer's comments, criticisms, and suggestions. This version of the article is more comprehensive, detailed, understandable, and contributes to the literature. Best regards.
Author Response

(The authors gave the same response as above.)

Reviewer 3 Report
Comments and Suggestions for Authors
I am very pleased with the modifications and recomand the article for publication.
Author Response

(The authors gave the same response as above.)
